# Direct Costs of Care for Adults with Soft Tissue Sarcomas: A Population-Based Study

**DOI:** 10.3390/cancers14133109

**Published:** 2022-06-24

**Authors:** Massimo Rugge, Alessandra Buja, Saveria Tropea, Giovanni Girardi, Luigi Cosenza Franzese, Claudia Cozzolino, Manuel Zorzi, Antonella Vecchiato, Paolo Del Fiore, Antonella Brunello, Alessandra Rosalba Brazzale, Vincenzo Baldo, Angelo Paolo dei Tos, Carlo Riccardo Rossi, Simone Mocellin

**Affiliations:** 1Pathology and Cytopathology Unit, Department of Medicine—DIMED, University of Padua, 35121 Padua, Italy; massimo.rugge@unipd.it (M.R.); angelo.deitos@unipd.it (A.P.d.T.); 2Veneto Tumor Registry, Azienda Zero, 35132 Padua, Italy; manuel.zorzi@azero.veneto.it; 3Department of Cardiologic, Vascular and Thoracic Sciences, and Public Health, University of Padua, 35131 Padua, Italy; giovanni.girardi.1@studenti.unipd.it (G.G.); luigi.franzese@gmail.com (L.C.F.); vincenzo.baldo@unipd.it (V.B.); 4Soft-Tissue, Peritoneum and Melanoma Surgical Oncology Unit, Veneto Institute of Oncology IOV-IRCCS, 35128 Padua, Italy; saveria.tropea@iov.veneto.it (S.T.); claudia.cozzolino@iov.veneto.it (C.C.); antonella.vecchiato@iov.veneto.it (A.V.); paolo.delfiore@iov.veneto.it (P.D.F.); simone.mocellin@iov.veneto.it (S.M.); 5Medical Oncology 1 Unit, Veneto Institute of Oncology IOV-IRCCS, 35128 Padua, Italy; antonella.brunello@iov.veneto.it; 6Department of Statistical Sciences, University of Padua, 35121 Padua, Italy; alessandra.brazzale@unipd.it; 7Department of Surgery, Oncology and Gastroenterology—DISCOG, University of Padua, 35128 Padua, Italy; carlor.rossi@unipd.it

**Keywords:** soft tissue sarcoma, cost of illness, economic impact, real-world data

## Abstract

**Simple Summary:**

This observational study provides a useful estimate of the cost of clinical pathways for soft tissue sarcoma at population level. Analyzing the drivers of the costs of a given illness emerging from these data could support the development of dynamic models for assessing cost-effectiveness based on real-world observations.

**Abstract:**

The clinical treatment of soft tissue sarcoma (STS) has evolved substantially over the last decade. This population-based cohort study based on real-world data included all incidental STS recorded by the Veneto Cancer Registry in 2017. Data on hospital admissions, emergency department and outpatient visits, drug prescriptions, and use of medical devices within two years from STS diagnosis were obtained from administrative databases. The average per-patient real-world costs over this two-year period, in total and by single expenditure item, were calculated and stratified by stage of disease at diagnosis, tumor histology and tumor site. The mean total cost per patient amounted to EUR 16,793. A higher TNM stage at diagnosis was associated with higher healthcare costs, as follows: compared with stage I, the average total cost per patient was 1.32, 2.18 and 3.36 times greater for stages II, III and IV, respectively. Hospital stays generated the greatest costs (averaging EUR 7950 per patient), followed by outpatient visits (mean EUR 3947 per patient) and drug prescriptions (mean EUR 3664 per patient). Given the paucity of population-based studies, the present results can serve as a reference for further cost-effectiveness analyses on care strategies for patients with STS.

## 1. Introduction

The incidence of soft tissue sarcomas (STS) in adults accounts for no more than 2% of all malignant tumors. STS include a spectrum of malignancies [1,2] that are classified by their (putative) cell lineage of differentiation as follows: adipocytic; fibroblastic/myofibroblastic; fibro-histiocytic; smooth-muscle; pericytic; skeletal-muscle; vascular; chondro-osseous; nerve sheath tumors; tumors of uncertain differentiation; or undifferentiated/unclassified sarcoma [3,4]. Within these major lineages-classes, more than 100 subtypes can be distinguished according to histological phenotype, immunohistochemical pattern, and molecular profile, and they differ significantly in terms of their clinical outcome. In Western countries, the mean five-year overall survival rate for patients with STS is around 65%, but it ranges greatly (from 80% to 15%) for different histotypes, neoplastic stages, and surgical sites [5].

The criteria for diagnosing and classifying STS have been significantly revised over the past twenty years in an effort to arrive at patient-centered (innovative) therapeutic strategies. Pathologists, radiologists, and surgical and medical oncologists have all been involved in tailoring new therapies to the biology and stage of neoplastic diseases [1,2,6]. While this multidisciplinary approach to patient care is expected to achieve significant prognostic advantages, the complexities of combining different clinical skills and interventions severely hinders any reliable estimation of the costs of diagnosing and treating a given disease.

To promote evidence-based clinical strategies, international agencies and scientific societies in Europe [7,8] and the USA [9] have developed different clinical practice guidelines (CPGs) for managing STS. The rate of adherence to these guidelines remains unsatisfactory, however (partly due to the changing diagnostic criteria), resulting in a significant variability in how the disease is diagnosed and treated, and also making it difficult to estimate the costs of care for patients with STS. [10,11,12] In an effort to support the best care strategies and the most rational allocation of resources, the Veneto Oncology Network (ROV) formally proposed standard diagnostic and therapeutic procedures to be implemented in cancer care units throughout the region. [13] The aim of the present population-based study was to estimate the direct costs sustained by the Veneto’s regional healthcare system for the care of adults with STS in the first two years after their diagnosis. Estimates were stratified by primary tumor site, histological STS cell lineage, and stage of the malignancy at presentation.

## 2. Materials and Methods

### 2.1. Context

The Italian National Health System is a public service supported mainly by general taxation, and organized on a regional basis. Its fundamental values are universality, free access, freedom of choice, pluralism in provision, and equity. The Veneto region in north-east Italy has a population of 4,908,000 (year 2017). Its cancer registration process covers all of the region’s resident population, and its registration procedures have been certified by the nationally recognized Bureau Veritas Certification Agency (2015; ISO 9001:2015).

In 2015, the Veneto Oncology Network (ROV) published a comprehensive document detailing the clinical procedures implemented at all stages of the clinical management of cases of STS, from the initial diagnostic workup to patients’ end-of-life care [13]. The ROV’s recommendations were based on the latest national and international literature, and included a detailed set of indicators to consider in monitoring the consistency between the proposed clinical management of STS and real-world clinical practice.

### 2.2. Patients’ Data

This population-based cohort study on real-world data considers all incident cases of STS recorded by the Veneto Cancer Registry (RTV) in the year 2017. Cancer registration was based on the classification of cases of STS by cell lineage, as in the latest available WHO Classification of Tumours of Soft Tissue and Bone—4th edition; 2013 [1].

The costs were estimated considering only the incident cases of STS in 2017, as recorded by the regional cancer registry. The estimated costs of care associated with a given histological lineage refer only to subtypes of STS accounting for more than five incident cases (adipocytic; fibroblastic/myofibroblastic; smooth muscle; vascular; of uncertain lineage differentiation). All remaining cases—including variants with a low incidence, or NOS STS (11/190)—were merged together in a miscellaneous group comprising the following: NOS STS (3 cases); peripheral nerve sheath STS (4 cases); undifferentiated small round cell STS (1 case); skeletal muscle tumors (1 case); so-called fibro-histiocytic STS (1 case); and chondro-osseous STS (1 case).

TNM staging of the cases of STS at the time of their initial diagnosis was performed according to the criteria established by the 8th edition of the AJCC (American Joint Committee on Cancer) [14].

### 2.3. Cost Analysis

The cost analysis was conducted from a health system perspective. Data on visits to outpatient clinics, specialist services, drug prescriptions, hospital or hospice admissions, treatments at the emergency department, and the use of medical devices were obtained from the regional administrative subject-level databases (see below). The cost of any diagnostic, therapeutic (surgical or other) interventions was based on the reimbursement rates established by the Veneto Regional Authority. For the cost assessment we considered the following sources:The Outpatient database, which contains information on all medical procedures (specialist visits, laboratory and radiological tests, radiotherapy and chemotherapy sessions, etc.) delivered at outpatient facilities under NHS funding, valued at the rate stated in the Tariff Nomenclature for outpatient services (TNOS), a detailed formulary of medical procedures for outpatients [15];The Hospital Admissions database, which includes the diagnosis-related group (DRG) for each admission, valued at the rate indicated in the Tariff Nomenclature for inpatient services (TNIP), a formulary covering all hospital activities, including day hospital admissions [16];The regional databases of outpatient drug prescriptions and in-hospital drug consumption, which records the costs of all medical therapies (including their dosage);The Emergency Department Admissions database, which records the cost of each admission, as the sum of all medical procedures undertaken;The Medical Devices database, which lists the costs of all medical devices reimbursed by NHS: tailored devices, disposable devices and medical aids for rare diseases [17];The Hospice database, which recorded the admission length of stay.

Each patient was linked via an anonymous unique identification code to all administrative data (see databases as reported below). All costs sustained over two years after STS was diagnosed were included. The average real-world costs per patient (total and by single item of expenditure) were calculated and stratified by the following clinical variables: site of primary tumor; histological phenotype; and TNM stage at initial cancer assessment.

Finally, a polynomial regression model (degree = 2) was developed to assess the trend of the survival-weighted and unweighted costs for higher TNM stages. The mean total survival-weighted costs were calculated by summing the average cost at first year plus the average cost at second year calculated including only those patients survived at first year.

All costs were calculated in euros. The data analysis and modelling were conducted using Python 3.8.8.

### 2.4. Ethics

To ensure confidentiality and anonymity, the Veneto Regional Authority removes all direct identifiers and replaces them with a code number in all datasets to retain the opportunity to link data from different administrative databases. The data analysis was performed using anonymous aggregated data with no chance of individuals being identifiable. Ethical approval for the study was obtained from the Veneto Oncological Institute’s Ethics Committee (n. 0001218/22).

## 3. Results

In 2017, the Veneto Cancer Registry recorded 197 cases of STS. After excluding 7 pediatric STS patients, this study only considered 190 incidental STS cases occurring in the region’s adult population (M: 106; F: 84; mean age 63.6; SD 16.0; median 64.5). Table 1 shows the patients’ distribution by age group, primary site of STS, cell lineage, and TNM stage at the initial diagnosis.

Table 2 shows the total mean and median per-patient costs over the first and the second year after STS diagnosis grouping by primary tumor site; cell lineage; and TNM stage. For the whole STS population, the mean cost of care in the two years after diagnosis was EUR 16,793. Costs were generally higher in the first year than the second (EUR 12,130 versus EUR 5538).

Table 2 also shows the ratio of each cost to the lowest cost identified in our sample. Retroperitoneal STS was associated with the highest mean costs (EUR 20,280), and primary trunk sites with the lowest (EUR 10,945).

The mean cost was also affected by STS histology. Considering the STS cells’ differentiation, the highest costs were reported for STS with a smooth muscle lineage (EUR 24,432) and the miscellaneous group (EUR 30,338). Vascular and fibroblastic/myofibroblastic STS cases were associated with the lowest mean cost (EUR 11,591, and EUR 12,348, respectively).

Average total costs rose steadily with stage of disease as follows: for survival-weighted costs from EUR 9888 for stage I to EUR 53,601 for stage IV; and unweighted cost from EUR 9803 for stage I to EUR 32,983 for stage IV (Figure 1). Analyzing differences in average unweighted whole costs between those deceased in the two years from diagnosis with respect to those who survived, evidence of higher costs in the former emerged and were as follows: EUR 22,343 (95% C.I. 17,966–26,719) versus EUR 17,361 (95% C.I. 13,586–21,136) in stage III cases, EUR 75,738 (95% C.I. 15,452– 13,602) versus EUR 23,689 (95% C.I. 11,394– 35,983) in stage IV patients.

The mean and median per-patient costs by item and by primary cancer site, histology, and TNM stage are shown in Table 3. Overall, hospitalization generated 47% (EUR 7950 on 16,793) of the mean costs, followed by 24% for outpatient visits (EUR 3947 on 16,793), and 18% for inpatient drug consumption (EUR 3037 on 16,793). None of the other costs considered—including outpatients drug prescriptions, medical devices, emergency department visits, and hospice costs—exceeded EUR 1000 each, and together they accounted for no more than 10% of the estimated mean costs.

When primary tumor site was considered, retroperitoneal STS was associated with the highest costs of care for both hospitalizations (EUR 10,417), and inpatient drug consumption (EUR 5144). When the limb was the primary site, the costs were highest for both specialist visits (EUR 4469), and medical devices (EUR 1450). Cases of STS involving the trunk carried the highest costs for Emergency Department Admissions (EUR 1645) (Table 3). The smooth muscle lineage was unequivocally identified as the histological variant resulting in the highest costs of care. Compared with all the other histological variants, STS featuring smooth muscle differentiation generated the highest costs of drugs for inpatients and outpatients (EUR 7839 and EUR 899, respectively), medical devices (EUR 1859), specialist visits (EUR 4478), and hospitalizations (EUR 8919) (Table 3).

The stages of STS significantly influenced the mean costs of care, and stage-related differences were more evident when dichotomized as low (stage I-II) versus high stages (III-IV). This was consistently attributable to the costs of the following: outpatients’ drug prescriptions (stages I-II versus III-IV: EUR 1100 versus EUR 1470); inpatients’ drug consumption (stages I-II versus III-IV: EUR 1342 versus EUR 16,930); medical devices (stages I-II versus III-IV: EUR 624 versus EUR 3445); emergency department visits (stages I-II versus III-IV: EUR 304 versus EUR 489); specialist visits (stages I-II versus III-IV: EUR 6054 versus EUR 11,135); and hospitalizations (stages I-II versus III-IV: EUR 13,307 versus EUR 20,478) (Table 3).

## 4. Discussion

Over the last two decades, spending on care for patients with neoplastic diseases has risen faster than the incidence of cancer [18,19]. Few studies have addressed the costs of care for patients with STS, and there are very few population-based studies among them. [20,21] The present population-based study provides a comprehensive picture of the direct costs associated with the overall, real-world clinical management of STS. As expected, the expenditure was found to correlate closely with primary tumor site, and stage of disease. However, this study also aimed to provide more information on the cost of managing cases of STS with a view to optimize resource allocation and ultimately improve patient care.

### 4.1. Costs of Care by Primary Site of STS

STS located in the trunk was associated with the lowest mean costs, at EUR 10,945. Taking this figure as a reference, the cost associated with retroperitoneal STS was almost twice as high, at EUR 20,280. The greater difficulty of accessing the retroperitoneum for therapeutic purposes may partly explain this difference. The mean cost relating to a retroperitoneal primary site was double the cost generated by any other primary location for inpatients, and far higher for outpatients (EUR 5144 versus EUR 712, respectively). The hospitalization-related costs for primary retroperitoneal STS (EUR 10,417) were likewise consistently higher than those generated by any other subtype of STS, whereas they were always less significant for emergency room visits, and the hospice-related costs of retroperitoneal STS were lower than for any other STS subtype.

### 4.2. Costs of Care by STS Lineage

Only five STS lineages [8] were considered in this study, i.e., fibroblastic/myofibroblastic, adipocytic, smooth muscle, vascular, and uncertain differentiation, while STS with a low incidence were merged into a sixth miscellaneous group. Among the five histological phenotypes, fibroblastic/myofibroblastic and adipocytic STS were the most prevalent (26.8%, and 25.3%, respectively). Fibroblastic/myofibroblastic and vascular STS were associated with the lowest mean costs of care (EUR 12,348 and EUR 11,591, respectively). Smooth muscle STS (EUR 24,432) generated the highest mean costs (more than twice as high as for the subtype of STS costing the least), and it ranked higher than the other lineages for all the items of expenditure considered. The very high mean cost associated with the miscellaneous group (EUR 30,338) suggests that further studies are needed with larger STS cohorts to provide a more detailed cost analysis. It is also important to bear in mind that the new classification criteria introduced in 2020 may have affected the diagnostic-therapeutic strategies adopted for STS, and may plausibly result in significant changes from the currently calculated costs of care.

### 4.3. Costs of Care by Stage of Disease

In the cohort of patients considered here, 32.1% had STS in stage I, while stage IV was the least prevalent (14.7%). This distribution of patients by stage of disease makes it difficult to compare our findings with other studies, which focused more on advanced STS. The cost of managing a case of STS in stage IV was more than three times higher (EUR 32,983) than for stage I (EUR 9803). The single most significant driver of these costs was hospitalization, which did not differ significantly between stage III and stage IV STS (EUR 10,419 and EUR 10,059, respectively). For patients in stage I, the costs of hospitalization and inpatient drug consumption amounted to EUR 6017 and EUR 1106, respectively, while the corresponding costs for stage IV were EUR 10,059 and EUR 15,908, respectively. While many European and North American studies have already reported on the costs of care for high-stage STS [10,22], the present analysis quantified the significant difference in the proportion of these costs by stage of disease for the first time. Collapsing the costs of outpatient and inpatient drug consumption, the percentage of expenditure on drugs was found to be 17% for STS patients in stage I, and 50% for those in stage IV. This is consistent with the findings of the above-mentioned European and North American studies, which demonstrated a significant increase in the direct costs of care with disease progression, and the important impact of chemotherapy-related costs on the clinical management of advanced STS [10,22].

This study has its limitations, including the limited number of cases considered, which ruled out any cost assessment by single STS histological subtype. The retrospective study design also significantly limited any detailed analysis of the diagnostic costs generated specifically by imaging, by histological or immunohistochemical procedures, or the use of molecular biology. Moreover, the costs esteems were derived from reimbursement tariffs, which could differ from actual sustained costs. In addition, the analysis only focuses on the direct costs of care, without considering the indirect costs sustained by patients and society. A further limitation is that this study does not include an evaluation of the costs of health care services obtained outside the region. Lastly, the study sample was drawn from a single region, therefore, the results should be cautiously interpreted for other settings.

On the other hand, the most significant strength of the present study lies in its population-based design, which allowed for the significant biases that intrinsically affect the information achievable by single-center studies to be minimized [23,24,25,26].

## 5. Conclusions

In conclusion, this study associated STS involving the retroperitoneum or limbs with the highest costs of care and showed that higher stages of disease are significantly more costly to manage, due largely to adjuvant chemotherapy after surgery. Given the paucity of population-based studies, the present results can serve as a reference for further cost-effectiveness analyses on care strategies for patients with STS.

## Figures and Tables

**Figure 1 cancers-14-03109-f001:**
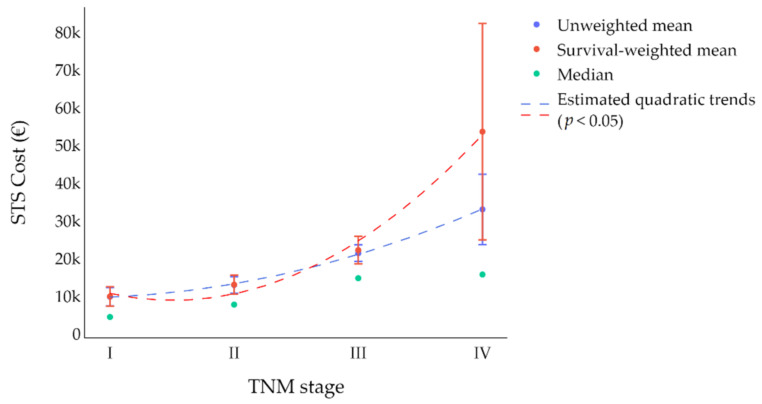
Survival-weighted and unweighted mean, median and polynomial regression estimates of costs of managing STS by TNM stage.

**Table 1 cancers-14-03109-t001:** Demographics and clinical variables of the sample.

Clinical Variable	Overall Number: 190 (%)	
**Sex**	Male	106 (55.8)
	Female	84 (44.2)
**Age**	20–29 (M:F = 3:5)	8 (4.2)
	30–39 (M:F = 2:0)	2 (1.1)
**Mean 63.6**	40–49 (M:F = 9:17)	26 (13.7)
**(SD: 16.0)**	50–59 (M:F = 18:23)	41 (21.6)
**Median 64.5**	60–69 (M:F = 25:11)	36 (18.9)
	70–79 (M:F = 27:17)	44 (23.2)
	80–89 (M:F = 19:9)	28 (14.7)
	≥90 (M:F = 3:2)	5 (2.6)
**Primary STS site**	Limbs	88 (46.3)
	Retroperitoneum	41 (21.6)
	Trunk	35 (18.4)
	Head-neck	19 (10.0)
	Unknown	7 (3.7)
**Lineage of differentiation**	Fibroblastic/myofibroblastic sarcoma	51 (26.8)
	Liposarcoma	48 (25.3)
	Uncertain differentiation	38(20.0)
	Leiomyosarcoma	37 (19.5)
	Vascular sarcoma	5 (2.6)
	Others	11 (5.8)
**TNM stage at initial** **diagnosis** **(VII AJCC Edition)**	I	61 (32.1)
	II	42 (22.1)
	III	51 (26.8)
	IV	28 (14.7)
	Unknown	8 (4.2)

**Table 2 cancers-14-03109-t002:** Costs (in EUR) at first, second year and survival- unweighted total costs.

Clinical Variable		Overall Number = 190	First Year Mean (Median)	Second Year Mean (Median)	Unweighted Total Costs into Two Years after Diagnosis			Ratio by Lowest	
					Mean (Median)	SD	[Min; Max]	Mean	Median
**Primary STS site**	Trunk	35	8998 (5143)	2433 (1615)	10,945 (9418)	8829	[50; 32,134]	-	-
	Head-neck	19	9943 (9777)	4316 (1457)	14,031 (12,107)	12,447	[160; 55,710]	1.28	1.29
	Limbs	88	11,965 (9874)	5527 (1719)	17,178 (12,647)	20,431	[0; 154,713]	1.57	1.34
	Retroperitoneum	41	14,016 (8838)	8561 (2017)	20,280 (11,293)	26,355	[1487; 163,571]	1.85	1.20
**Lineage of** **differentiation**	Vascular sarcoma	5	8421 (7270)	3962 (3572)	11,591 (8392)	6512	[3783; 20,791]	-	1.21
	Fibroblastic/myofibroblastic sarcoma	51	9243 (5496)	3300 (876)	12,348 (6924)	13,238	[0; 62,865]	1.07	-
	Liposarcoma	48	11,085 (7804)	3365 (1767)	14,099 (10,506)	12,02	[55; 45,463]	1.22	1.52
	Uncertain differentiation	38	12,645 (12,907)	4499 (1558)	15,487 (13,063)	12,327	[160; 58,488]	1.34	1.89
	Leiomyosarcoma	37	15,288 (9833)	10,914 (2220)	24,432 (12,412)	34,502	[0; 163,571]	2.11	1.79
	Others	11	19,350 (14,109)	12,086 (4329)	30,338 (19,121)	40,036	[6238; 154,713]	2.62	2.76
**TNM stage** **at initial** **diagnosis (VII AJCC Edition)**	I	61	8161 (4436)	1727 (649)	9803 (6238)	11,169	[0; 62,865]	-	-
	II	42	9454 (7733)	3559 (1597)	12,928 (11,927)	9506	[0; 39,917]	1.32	1.91
	III	51	15,319 (14,729)	6853 (2325)	21,366 (19,121)	13,248	[1189; 67,713]	2.18	3.43
	IV	28	21,529 (15,709)	32,072 (11,217)	32,983 (16,880)	44,559	[2047; 163,571]	3.36	2.71
**Total**		190	12,130 (9017)	5538 (1736)	16,793 (12,126)	21,676	[0; 163,570]		

**Table 3 cancers-14-03109-t003:** Survival- unweighted costs (in EUR) of specific items in the two years after diagnosis.

		Outpatient Drugs			Inpatient Drugs			Medical Devices		
Clinical Variable		Mean (Median)	SD	[Min; Max]	Mean (Median)	SD	[Min; Max]	Mean (Median)	SD	[Min; Max]
**Primary STS site**	Trunk	586 (240)	723	[0; 3493]	447 (24)	1056	[0; 5539]	188 (0)	990	[0; 5943]
	Head-neck	516 (341)	541	[0; 2147]	1252 (5)	4593	[0; 20,659]	1280 (0)	3879	[0; 15,877]
	Limbs	651 (295)	968	[0; 5543]	2543 (51)	14,525	[0; 132,613]	1450 (0)	6708	[0; 51,700]
	Retroperitoneum	712 (503)	813	[0; 4273]	5144 (83)	21,709	[0; 139,870]	528 (0)	1865	[0; 8762]
**Lineage of differentiation**	Vascular sarcoma	589 (254)	680	[0; 1892]	12 (0)	14	[0; 33]	112 (0)	163	[0; 420]
	Fibroblastic/myofibroblastic sarcoma	424 (241)	601	[0; 3195]	923 (15)	4781	[0; 34,367]	932 (0)	3861	[0; 22,915]
	Liposarcoma	523 (237)	603	[0; 2295]	1337 (65)	3519	[0; 18,240]	632 (0)	3691	[0; 25,777]
	Uncertain differentiation	624 (283)	989	[0; 5543]	563 (63)	1086	[0; 5539]	831 (0)	2932	[0; 15,877]
	Leiomyosarcoma	899 (664)	1028	[0; 4273]	7839 (111)	27,735	[0; 139,870]	1859 (0)	8527	[0; 51,700]
	Others	1127 (675)	1144	[9; 3900]	14,035 (478)	37,806	[0; 132,613]	143 (0)	399	[0; 1403]
**TNM stage at** **initial** **diagnosis (VII AJCC Edition)**	I	517 (175)	795	[0; 3493]	1106 (24)	4844	[0; 34,368]	44 (0)	207	[0; 1403]
	II	583 (430)	597	[0; 2290]	236 (23)	528	[0; 2072]	580 (0)	2685	[0; 15,877]
	III	845 (607)	1052	[0; 5543]	1022 (163)	2371	[0; 15,924]	2595 (0)	8671	[0; 51,700]
	IV	625 (267)	891	[0; 4273]	15,908 (261)	38,555	[0; 139,870]	750 (0)	2217	[0; 8762]
**Total**		627 (313)	849	[0; 5543]	3037 (36)	16,029	[0; 139,870]	949 (0)	4856	[0; 51,700]
		**Emergency Department Visits**					**Specialist Visits**			
**Clinical Variable**		**Mean** **(Median)**			**SD**	**[Min; Max]**	**Mean** **(Median)**		**SD**	**[Min; Max]**
**Primary STS site**	Trunk	1645 (23)			330	[0; 1652]	3739 (1497)		4406	[0; 15,815]
	Head-neck	217 (143)			258	[0; 823]	3641 (560)		4984	[0; 15,976]
	Limbs	170 (23)			293	[0; 1200]	4469 (2677)		4489	[0; 20,316]
	Retroperitoneum	249 (86)			410	[0; 2109]	3226 (2046)		4166	[0; 18,637]
**Lineage of differentiation**	Vascular sarcoma	144 (71)			157	[0; 397]	4437 (560)		5397	[0; 13,542]
	Fibroblastic/myofibroblastic sarcoma	165 (0)			333	[0; 1652]	2926 (1504)		4127	[0; 20,316]
	Liposarcoma	170 (0)			305	[0; 1415]	3486 (1620)		4030	[0; 13,303]
	Uncertain differentiation	254 (100)			293	[0; 970]	4206 (2853)		4137	[0; 12,367]
	Leiomyosarcoma	230 (94)			396	[0; 2109]	4478 (2677)		5428	[0; 23,419]
	Others	79 (23)			115	[0; 349]	7788 (8923)		5852	[29; 18,337]
**TNM stage** **at initial** **diagnosis (VII AJCC Edition)**	I	137 (0)			332	[0; 2109]	1982 (962)		3387	[0; 20,316]
	II	167 (12)			263	[0; 1034]	4072 (2633)		3821	[0; 15,815]
	III	283 (99)			400	[0; 1652]	6101 (4795)		4433	[134; 18,337]
	IV	206 (135)			216	[0; 686]	5034 (862)		6645	[0; 23,419]
**Total**		191 (45)			323	[0; 2109]	3947 (2152)		4674	[0; 23,419]
		**Hospitalization**					**Hospice**			
**Clinical Variable**		**Mean** **(Median)**			**SD**	**[Min; Max]**	**Mean (Median)**		**SD**	**[Min; Max]**
**Primary STS site**	Trunk	5724 (3702)			5203	[0; 18,400]	96 (0)		418	[0; 2310]
	Head-neck	7002 (6013)			5095	[0; 17,879]	122 (0)		369	[0; 1470]
	Limbs	7835 (5398)			7995	[0; 41,310]	60 (0)		369	[0; 3272]
	Retroperitoneum	10,417 (5948)			10,722	[0; 47,113]	4 (0)		28	[0; 182]
**Lineage of differentiation**	Vascular sarcoma	6003 (6468)			1733	[2715; 7547]	294 (0)		588	[0; 1470]
	Fibroblastic/myofibroblastic sarcoma	6958 (4088)			7086	[0; 33,122]	22 (0)		146	[0; 1050]
	Liposarcoma	7952 (4467)			7996	[0; 33,715]	0 (0)		0	[0; 0]
	Uncertain differentiation	8831 (6239)			9659	[0; 45,859]	178 (0)		617	[0; 3272]
	Leiomyosarcoma	8919 (7373)			8423	[0; 47,113]	209 (0)		804	[0; 4410]
	Others	7127 (6266)			6609	[0; 21,408]	38 (0)		121	[0; 420]
**TNM stage** **at initial** **diagnosis (VII AJCC** **Edition)**	I	6017 (3702)			7454	[0; 45,859]	0 (0)		0	[0; 0]
	II	7290 (5347)			6664	[0; 28,540]	0 (0)		0	[0; 0]
	III	10,419 (8535)			8308	[0; 41,310]	101 (0)		495	[0; 3272]
	IV	10,059 (7885)			9802	[0; 47,113]	401 (0)		964	[0; 4410]
**Total**		7950 (5398)			8,09	[0; 47,113]	92 (0)		476	[0; 4410]

## Data Availability

The data supporting the findings of this study are held by the Veneto Epidemiological Registry and were used under license for the present work, but they are not publicly available. These data are nonetheless available from Manuel Zorzi on reasonable request and subject to permission being obtained from the Veneto Epidemiological Registry (Veneto Regional Authority).

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
