# Peer review of "Direct Costs of Care for Adults with Soft Tissue Sarcomas: A Population-Based Study"

_cancers, 2022, doi:10.3390/cancers14133109_

Round 1

Reviewer 1 Report

The authors report the results of an observational study of the direct costs of care for patients with soft tissue sarcoma. The total cost per patient is mainly determined by the costs of hospitalization, the costs of specialist visits and the costs of the drugs prescribed. Additionally, patients with a severe diagnosis are associated with higher healthcare costs.

Information on the costs of care is often lacking and this paper add new data on this issue.

Some minor revisions are requested on paragraph 2.3 “cost analysis”:

- The authors need to clearly explain for each database which costs were extracted and which Tariffs were applied. From the outpatient database did you obtained only the costs of specialist visits? Did you include costs for the laboratory examination or the radiographic evaluation? In the costs of medical devices, which categories of devices were included?

- In Italy, the national database of medical devices includes information relating to contracts for the acquisition and consumption of medical devices. The reference unit of this database is the medical device. It may be a strength to explain how consumption of medical device per patients can be derived in your Region.

- Table 3c includes hospice cost. From which database was this cost obtained?

- How was total survival-weighted costs calculated?

Reviewer 2 Report

This manuscript is an innovative paper on costs in the treatment of STS.
The work needs linguistic revision to make it more fluent.
The limitations of the study have been acknowledged by the authors, but I recommend adding among the limitations that the analysis was done on a sample drawn from a single region and with a single pay system.

After this minor revision, the study will be worthy of publication due to its originality.

Author Response

The work needs linguistic revision to make it more fluent.

Answer: The English editing has been supervised by a mother tongue specialist medical writer. We thank the reviewer for her/his appropriate criticisms.

The limitations of the study have been acknowledged by the authors, but I recommend adding among the limitations that the analysis was done on a sample drawn from a single region and with a single pay system.

Answer: Appropriate implementation/s has/have been added to the R1-version (in yellow), including those related to the specific reviewer’s criticism.

After this minor revision, the study will be worthy of publication due to its originality.